# A New End-to-End Workflow for the Community Earth System Model (version 2.0) for CMIP6

Sheri Mickelson, Alice Bertini, Gary Strand, Kevin Paul, Eric Nienhouse, John Dennis, and Mariana Vertenstein

The National Center for Atmospheric Research, Boulder, CO, USA

**Correspondence:** Sheri Mickelson (mickelso@ucar.edu)

**Abstract.** The complexity of each Coupled Model Intercomparison Project grows with every new generation. The Phase 5 effort saw a dramatic increase in the number of experiments that were performed and the number of variables that were requested compared to its previous generation, Phase 3. The large increase in data volume stressed the resources of several centers including at the National Center for Atmospheric Research. During Phase 5, we missed several deadlines and we struggled to get the data out to the community for analysis. In preparation for the current generation, Phase 6, we examined the weaknesses in our workflow and addressed the performance issues with new software tools. Through this investment, we were able to publish approximately 565 TB of compressed data to the community, with another 30 TB yet to be published. When compared to the volumes we produced in the previous generation, 165 TB of uncompressed data, we were able to provide six times the amount of data and we accomplish this within one-third of the time. This provided us an approximate 18 times speedup. While this paper discusses the improvements we have made to our own workflow for CMIP6, we hope to encourage other centers to evaluate and invest in their own workflows in order to be successful in these types of modeling campaigns.

## 1 Introduction

The Coupled Model Intercomparison Project Phase 6 (CMIP6) (Eyring et al., 2016) is a large international project that consists of many centers around the world running the same simulations, in order to seek a better understanding of Earth processes under different scenarios. This includes, but not limited to, studying different mitigation strategies, paleo climate analysis, and different land mitigation strategies. Centers commit to running a core (or DECK) set of experiments along with different tiers of experiments that can be compared against the DECK experiments. The National Center for Atmospheric Research (NCAR) committed to running most tier 1 experiments from almost all of the different Model Intercomparison Project (MIP) groups. In total, this included running 130 unique experiments with many having multiple ensemble members. This commitment required over one thousand different model runs, simulating over 37,000 years of climate. This consumed over 190 million CPU hours and produced over 2 PB of model output time-series data and 600 TB of requested formatted data.

During the Coupled Model Intercomparison Project Phase 5 (CMIP5) (Taylor et al., 2012), the post-processing of the data became a large problem for NCAR. During that process, NCAR used the Community Earth System Model (CESM) version 1 (Hurrell et al., 2013) to generate roughly 2.5 PB of raw output in about 18 months. It then took NCAR an additional 18 months

to post-process and publish the data. Due to inefficiencies in both the post-processing software and workflow orchestration, NCAR was only able to publish about 165 TB of data. To help ease the process of running the CMIP6 experiments and post-processing the data, NCAR invested resources to improve the scientific workflow to ensure everything would be published to the community efficiently. These changes were required to work with the new version of the model, the Community Earth System Model version 2 (Danabasoglu et al., 2020), to be as efficient as possible, and they needed to reduce the human burdens caused by running such experiments.

In order to improve our end to end workflow, we needed to focus on three areas. The first step was to improve the performance of the data workflow by creating a set of new tools that would allow us to parallelize each of the operations and streamline the publication process. This work is discussed in Section 2. Second, we needed to automate the process workflow in order to remove the expertise needed to run the different tasks and to have tasks run continuously without intervention. This is discussed in Section 3. Finally, we needed a better way to track simulation progress and document the experiments. The improvements that were made in this area are discussed within Section 4.

## 2 Data Workflow

The first task in creating a new data workflow for CMIP6 was to evaluate the methods used in CMIP5 in order to find where improvements needed to be made. The life cycle of the data consists of the multiple stages shown in Fig. 1. First, the model is ran and raw model output is generated. As the model runs, diagnostics are generated in order to track the simulation's scientific progress. For CMIP5, this was a manual process that was not often done because it would take several hours for users to setup and run a full set. When the model run is complete, the raw output is transformed into a time-series format. For CMIP5, this process did not contain any parallelism and it was slow to run because of the amount of data that was required to be post-processed. The time-series formatted data are then used to generate a new set of data that complies to the specific MIP standards that are defined within Taylor et al. (2017) and within the CMIP6 data request (Juckes et al., 2020). For CMIP5, this process also did not contain any parallelism and it was slow to run because of the amount of data that was required to be post-processed. In addition, our software for this process was difficult to run as it required expert knowledge to ensure the data that was generated met the correct MIP standards. After the standardized data were verified by the scientist, it was then published to the Earth System Grid Federation (ESGF) (Cinquini et al., 2014).

Fundamentally, the post-processing steps involved opening a set of files and reading the data, performing one or more simple operations on the data, and then writing out the results. While the post-processing steps were straight forward, they were very time consuming to run due to the number of files and total data volume on which they operated. For example during CMIP5, which had data volumes in the several tera-byte range, post-processing calculations would take several days to run for each experiment. For CMIP6, which involved a significantly larger number of files and total data volumes in the peta-byte range, a better solution was needed. In particular we needed tools with flexible interfaces, that could write compressed NetCDF files in parallel, and minimized the number of times output files were opened and closed for writing.

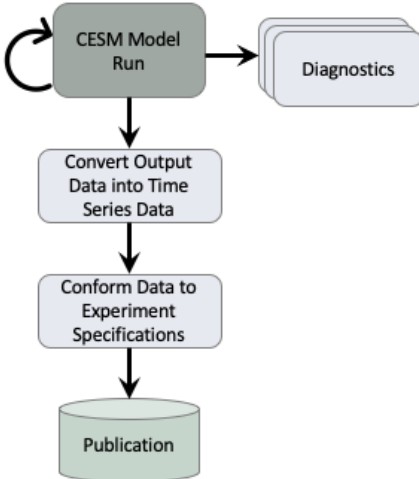

**Figure 1.** This flowchart describes the tasks that are executed within the CESM workflow in order to generate data for CMIP. The diagnostics task can be executed several times while the CESM model run is executed. The remainder of the tasks are each executed once when the CESM model run completes. This flow chart is a simplistic view of our general workflows. In practice, our workflows are more similar to the workflow depicted in Appendix Fig. A1.

There are a number of existing software package that can be used to perform the post-processing steps including: the NetCDF Operators (NCO) (Zender, 2008), the Ultrascale Visualization Climate Data Analysis Tools (UVCDAT) (Williams, 2014), the Climate Data Operators (CDO) (Kornblueh et al., 2019), and Pagoda (Daily, 2013). While these packages provide a diverse set of operations, none of them satisfied all of the necessary requirements. For example while CDO minimized the number of times output files were opened and closed, it did not easily enable parallel execution. Conversely while Pagoda offered parallel execution, it did not minimize the number of openings and closings. The XML IO Server (XIOS) (Meurdesoif, 2020) is an IO library that is able to write publication ready output directly from the model. While XIOS provides excellent performance, implementing this method would have required us to rewrite the IO interface within all of the modeling components and this would have required more people to work on this option than were allotted for this project. We therefore decided to develop our own tools based on Python and the Message Passing Interface library (MPI) (Gropp et al., 1999) to enable parallelism. We choose to use Python because of its flexibility, available libraries, and quick prototyping ability (Perez et al., 2011; Oliphant, 2007) and MPI4Py (Dalcin, 2019) library to enable parallelism. These benefits of Python allowed two full-time employees to create the post-processing tools presented in this section within the three year timeline we needed them completed by.

We also saw performance issues during the publication of CMIP5 data contributions to the ESGF. During CMIP5, the ESGF software stack was stressed when large amounts of data were trying to be published by multiple organizations at the same time. Over the past few years, a team of individuals from around the world have been improving the ESGF software stack (Abdulla,

2019). The process improvements that were made to ESGF, along with the post-processing tools we developed are described in the following subsections.

Most of the performance improvements that are described in the following subsections were ran on the Cheyenne super-computer (Cheyenne, 2017). Cheyenne is a 5.34-petaflop machine that contains 4,032 dual-socket nodes. Each node contains two Intel Broadwell processors that are clocked at 2.3-Ghz. For these tests, we used the standard nodes that contain 64 GB of memory per node.

We also provide timing results from the Yellowstone supercomputer (Yellowstone, 2017). Yellowstone is a 1.51-petaflop machine that contains 4,536 nodes, each containing dual Intel Sandy Bridge processors. Each node contains 32 GB of memory.

## 2.1 Time Series Generation

The first step within our post-processing workflow involved a transformation of the raw CESM output data from time slice into time series. This operation is represented in the "Convert Output Data into Time Series Data" task within Fig. 1. Each of the CESM components produces output files that contain multiple variables in one time slice chunk. Unfortunately this is not an ideal format for distribution, because scientists are typically interested in evaluating a handful of variables at multiple time steps. In order to increase the usability of the data, the data are reformatted into a time-series format, where each file contains one or more time slices of a single variable.

In addition to transforming the data, this process also needed to verify that all time slices were inserted correctly into each time series file. This involved sorting all of the time slices, verifying that the time values were all unique, ensuring there were no gaps in the time dimension, and correctly inserting the time slices into chronological order.

Interestingly, the conversion of time-slice to time-series data was the single most expensive component of the CMIP5 workflow. While this operation is embarrassingly parallel due to the lack of data dependencies between each variable, the serial CMIP5 workflow used individual NCO commands that opened, read and wrote each individual time slice. Consider the number of file operations necessary to convert an entire data-set which contains $num_{TS}$ (number of time slices) and $num_{var}$ (number of variables) from time-slice to time-series. Using the serial CMIP5 workflow, the execution consisted of $2 \times num_{TS} \times num_{var}$ open and close operations and $num_{TS} \times num_{var}$ read and write operations. We were able to significantly reduce the number of these expensive disk I/O operations through the creation the PyReshaper (Paul et al., 2015, 2018). We next describe the PyReshaper tool which was adopted into the CESM post-processing framework (Bertini and Mickelson, 2019)

The approach used by PyReshaper is illustrated in Fig. 2. An MPI rank is assigned one or more fields to read from the time-slice file and write to the time-series file. Each MPI rank $i$ operates independently and performs $num_{var}^i * num_{TS} + 1$ open and close operations and $num_{var}^i * num_{TS} + 1$ read and write operations where $num_{var}^i$ is the number of fields assigned to MPI rank $i$. Given a sufficient amount of memory, it is possible to further reduce the number of write operations by writing multiple time slices to the file system in a single call. This task based parallelism supports execution on as many MPI ranks as there are fields in the input data set. Ideally if all the input fields were the same size and the cost to read the data from and write the data to the file system was negligible it would be possible to achieve a maximum speedup of $num_{var}$. Unfortunately

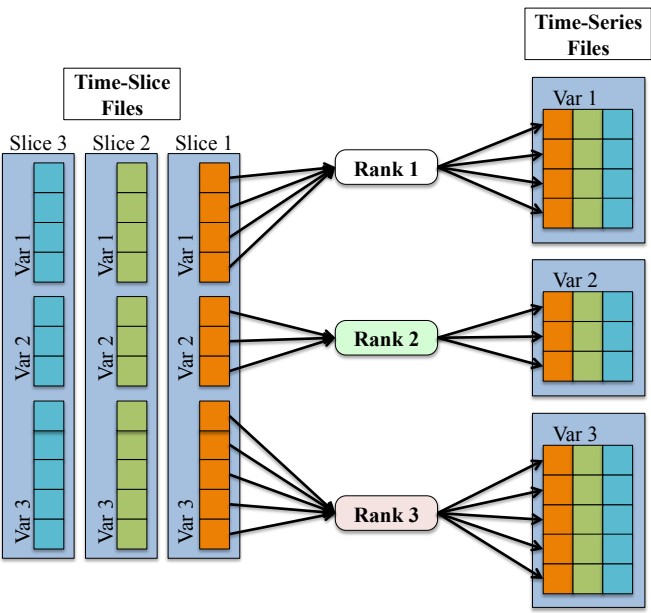

**Figure 2.** This figure shows the process of converting the data from a time-slice format to a time-series format in parallel within the PyReshaper. Each MPI rank is responsible for taking a particular variable from each time-slice file and writing it to the time-series file.

the size of all input fields are not the same and the cost of read data from and write data to the file system is not negligible. We next describe the actual speedup the PyReshaper approach enables.

In the performance evaluation of PyReshaper, we evaluated the time it took to convert 10 years of monthly atmospheric data into the time-series format. This test configuration represents the conversion of approximately 180 GBytes of input data. The conversion took approximately 5 ½ hours using the existing serial method on NCAR's Cheyenne supercomputer.

Figure 3 illustrates the performance improvements of the PyReshaper tool over the existing method. Note that using 144 MPI ranks we achieve the same conversion in approximately 4 ½ minutes.

The large improvement seen between 144 ranks and 72 ranks is an indication of a load-imbalance in the partitioning of fields to MPI ranks within the PyReshaper tool. This behavior occurs because the algorithm does not take into account any difference in processing cost between variables. Therefore, some ranks can end up with more expensive three-dimensional variables to process while others may get only two-dimensional variables.

For this evaluation we did not scale above 144 ranks because this is what we had run in production. As noted above, the PyReshaper does not attempt to load balance between the ranks and as the PyReshaper completes, several ranks remain idle while others still complete their work. We found 144 ranks to be a good balance of resources per throughput based on the average number of three-dimensional and two-dimensional variables, which was verified by running throughput tests.

Future work will be needed to handle the load-balancing within the PyReshaper. Load-balancing techniques that could be implemented include a coordinator-worker task assignment method. Another naive implementation would involve assigning

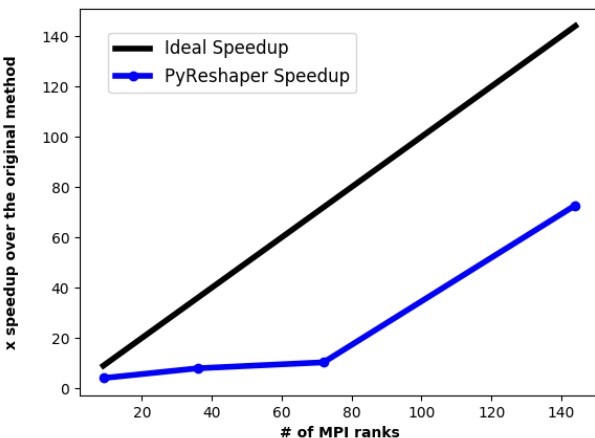

**Figure 3.** The comparative speedup in time of creating time-series files from ten years of monthly atmospheric data. In all cases, 493 time-series variable files were created. For comparison, the previous CMIP5 sequential methods took approximately 5 ½ hours to complete. With 144 MPI ranks we were able to bring the time to do this same conversion down to approximately 4 ½ minutes.

work based on evenly dividing the three-dimensional and two-dimensional variables amongst the ranks. Either method would create a more predictable scaling that would reduce the need to study performance tests based on different problem sizes in order to achieve desired performance.

## 2.2 Diagnostics

One of the main ways the NCAR scientists evaluate the output of CESM during modeling campaigns such as CMIP5 and CMIP6, is to run the component diagnostic packages. This task is represented by the "Diagnostics" task within Fig. 1. They consist of four separate packages which are used to evaluate atmosphere, ocean, ice, and land model output. Each of these packages used a combination of shell scripts, NCO, and NCL (NCL, 2019) to create a set of average files, or climatology files, plotted the data against observations or another model run, and then created an HTML document that linked all of the plot image files. While NCL was the preferred language to create these plots, with a few modifications, any of the packages could create plots in other languages. The HTML documents generated from our diagnostic packages can be found off of our landing page (CESM Diagnostics Results, 2019) and an example set of diagnostics specifically from one of our CMIP6/PMIP4 experiments can be found within these links (Atmospheric Diagnostics Results, 2019; Ocean Diagnostics Results, 2019; Land Diagnostics Results, 2019; Sea Ice Diagnostics Results, 2019).

Each package requires different types of climatologies and plot types which creates unique performance characteristics for each of the packages. While previous efforts have enabled parallelism in the diagnostic packages (Woitaszek et al., 2011; Jacob et al., 2012), this approach resulted in poor performance for multiple file operations, and it had a steep learning curve for users. In order to create the climatology files in parallel and to reduce the expensive disk I/O operations, we developed the

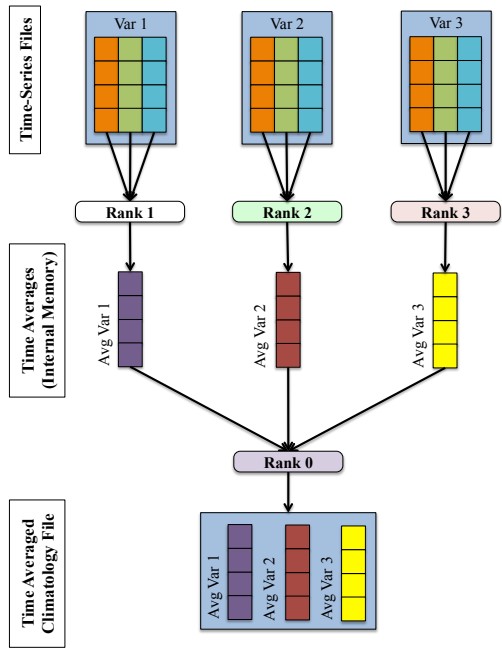

**Figure 4.** A depiction of the parallelism strategy that the PyAverager uses for writing each climatology file. This figure describes how four MPI ranks are tasked within an example subcommunicator.

tool PyAverager (Paul et al., 2015; Mickelson et al., 2018). We also chose to call the NCL plotting scripts in parallel in order to improve performance further.

The parallelism strategy the PyAverager uses is illustrated in Fig. 4. When the application begins, the pool of MPI ranks are partitioned into subcommunicators and the climatologies to be computed are partitioned across all subcommunicators. One MPI rank in each subcommunicator is assigned to be the writer of the given climatology file. Then, the field list is partitioned across the remainder of MPI ranks within the subcommunicator. Each of these ranks is responsible for retrieving its assigned field, computing the correct climatology, and then sending the result to the writer. After all fields have been written, the subcommunicator group begins computing the next climatology file it was assigned.

The number of MPI ranks within a subcommunicator was set to four. If the total number of MPI ranks that were given to the PyAverager was less than four or there were less than four variables that needed to be operated on, the number of ranks within a subcommunicator was set to two. The total number of subcommunicators was computed by dividing the total of MPI ranks by the number of ranks within a subcommunicator. Once the MPI ranks were evenly distributed to their corresponding subcommunicators, the averages where then assigned evenly amongst the subcommunicators.

The second part of the diagnostics involves creating plots from the climatologies that were created. The plotting scripts individually can take a long time to run and run times vary among the plotting scripts. In order to improve the performance further, the CESM post-processing framework calls the existing individual NCL scripts and some newly created Python plotting

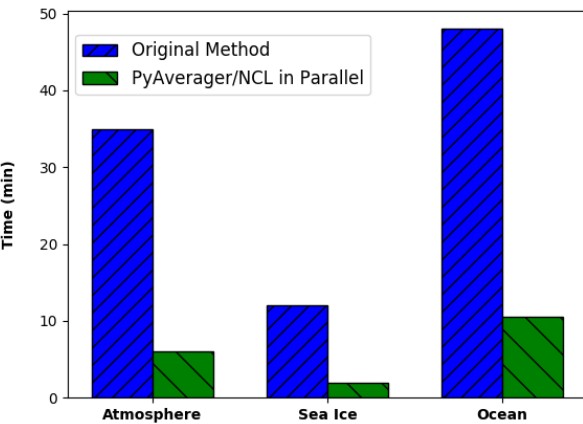

**Figure 5.** The performance comparison across different diagnostic packages from 10 years of monthly CESM data. These timings include the total time to create all of the required climatology files and to run each of the NCL plotting scripts. The PyAverager/NCL in Parallel timings were all computed using 16 MPI ranks.

scripts in parallel. We are able to execute them in parallel because there are no data dependencies within the scripts. Therefore, if we have as many MPI ranks available as we do plotting scripts, the performance is limited to the longest running script.

In order to evaluate the performance of our improvements, we ran original versions of the diagnostic packages and compared them to the time it took our new version to create the same climatology files and the same NCL plots. We ran these comparisons on the Yellowstone supercomputer and we used 16 MPI ranks for all PyAverager/NCL in parallel timings. Figure 5 shows that we were able to achieve a 5.8 times speedup for the atmospheric diagnostics, a 6 times speedup for the ice diagnostics, and a 4.6 times speedup for the ocean diagnostics.

In order to evaluate the scalability of the PyAverager, we compared the time it took to create twelve monthly and four seasonal climatology files with the PyAverager against the NCO tools ran in serial. We chose to operate on the same data that was used to evaluate the performance of the PyReshaper in the previous section and all timings were performed on Cheyenne.

You can see from Fig. 6 that the PyAverager is able to scale better than the PyReshaper. This is because the problem size is more load balanced. As you recall, the PyAverager distributes the number of averages to be done amongst the available subcommunicators and the number of variables are distributed amongst the ranks within the subcommunicator. For this particular problem, the work is more evenly distributed because the problem sizes were all similar and this lead to the better scaling.

The lack of improvement seen between ranks 16 and 32 is because the work wasn't evenly distributed and a subcommunicator ended up with slightly more work to do. This was unavoidable because of the order in which the tasks were assigned. To improve the performance on 32 tasks, we would have to evaluate the problem size before assignment and evenly distribute the work among the subcommunicators. This can be difficult to predict because some calculations can become more expensive

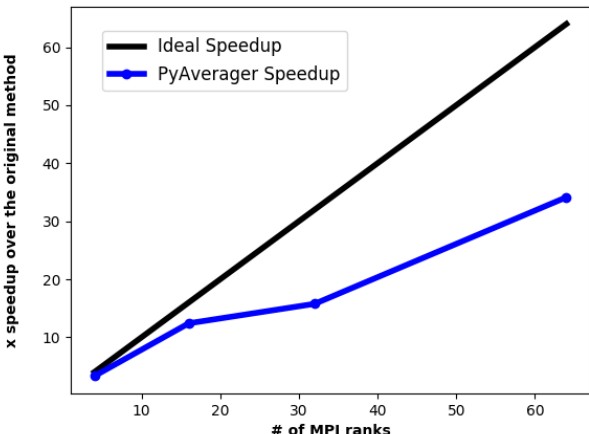

**Figure 6.** The comparative speedup of creating climatology files from ten years of monthly atmospheric data. Four seasonal and twelve monthly climatology files were created. For comparison, the original methods took approximately 26 ½ minutes to generate the climatologies. The PyAverager took approximately 46 seconds to create the same climatologies with 64 MPI ranks.

under different variable sizes. We chose to avoid this complexity because we were content with the improvements we had seen, but this is something we can improve on in the future.

## 2.3 Conforming Data to Meet Specifications

The final step before publishing the data involves conforming the data to meet experiment specifications. This is represented as the "Conform Data to Experiment Specifications" task within Fig. 1. This requirement is done in order to enable scientists to directly compare the data from different centers without having to perform data transformations that can be error prone. Some examples include renaming model variables, combining fields (e.g., adding or subtracting) to create one output field, converting units, verifying the data resides on the specified grid, and checking that the correct attributes are attached to the files. The recommended method to create the specified data requires users to write code to make required data transformations and to call the Climate Model Output Rewriter (CMOR) (Taylor et al., 2006) library to check for compliance and to add file attributes. The software we had written and used for CMIP5 used this recommended method, but it was written as serial code and it took a long time to execute on a large data set. It was also difficult to extend this software to include the many additional variables added for CMIP6. In order to meet the demands of CMIP6, we developed the tool PyConform (Paul et al., 2016, 2019) because we needed a tool with a flexible interface that could adapt to changes in requirements more easily, that could create variable output in parallel, and still produced data that met specification requirements.

An example of a PyConform job is shown in Fig. 7. The input fields are found on the left side of the figure. These fields are operated on as they are fed through the system in order to produce the output fields on the right. There are a variety of operations that can be performed on the data and this figure only shows a small subset. Several common functions and

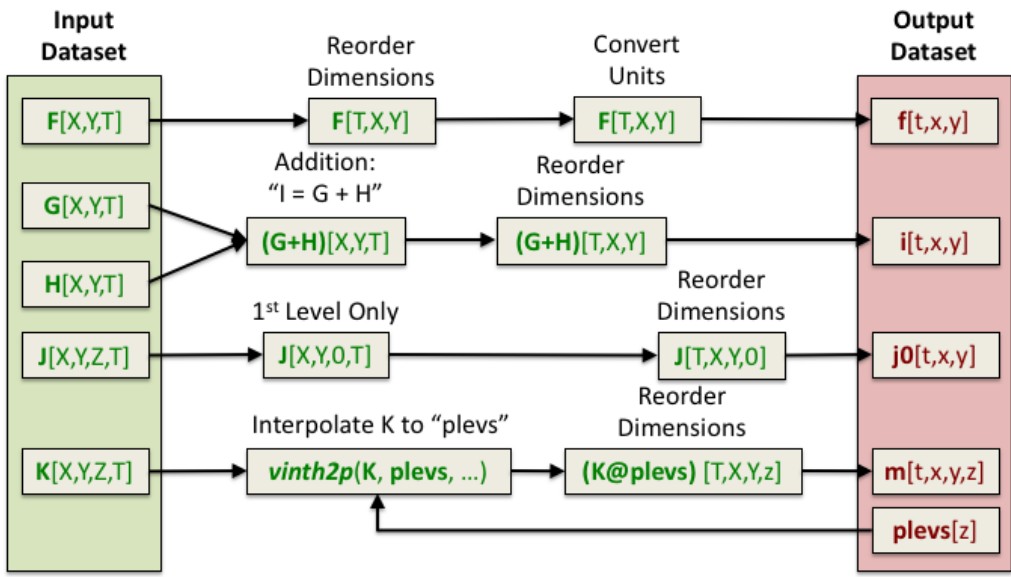

**Figure 7.** An example of a PyConform job. Each MPI rank is responsible for creating a particular output data set. Its job is to retrieve the variable data it needs, map operations, execute these operations, and then write the data.

arithmetic operations are provided with the tool, but we could not account for all functions users may need. We provide an example PyConform CESM model output to MIP compliance definition file (Mickelson, 2019a) to list the available functions

and operations that PyConform provides. If more functionality is needed, we provide a framework in which users can create their own functions in Python and plug them into the framework. For this application, we again relied on a task-parallel approach in which an MPI rank was assigned to create one output file. Once the file was written, the MPI rank was given another output file to create.

PyConform depends on the CMIP6 data request Python API, `dreqPy` (Juckes et al., 2020). This package interfaces with

200 the CMIP6 data request database which contains information regarding all of the fields within the request. This includes field names, descriptions, units, coordinates, and other specific information. Experiment information is also contained within the data request, specifying experiment descriptions and which fields are being requested for that experiment.

During the development of PyConform, we chose to keep our `dreqPy` interface code as flexible as possible. `dreqPy` was intended to be an evolving database, adding new fields and experiments in time, and PyConform needed to be able to handle

new information without any code modifications. Once the user installed the latest verion of `dreqPy` in their path, PyConform automatically queries the `dreqPy` package to obtain experiment and field information. This information is then used within the PyConform software to generate the requested field output files with the correct attributes attached to it.

Flexibility was also needed within the interface used to define how CESM data would be used to derive the variables that were being requested for CMIP6. We chose to use text files (Mickelson, 2019a) to define how these variables would be created.

The variable definitions within the text files follow the simple format `cmipvariable=modelvariable`. These variable

definitions were provided by and verified by many of the scientists who work on the CESM model. If we needed to make any modifications due to changes within the CESM model, changes within the CMIP6 data request, or if a variable was added, all that was required was to add a line to the text file or modify a line. This allowed us to make modifications quickly because we did not have to modify any Python code. Instead PyConform would see the updated information in the text definition file and automatically use the new definition to create the output file.

The flexibility we added in for the PyConform tool allowed us to fix data quickly if inconsistencies were found. Once we retracted data, we where able to republish data within a few days because we were able to make modifications quickly to a text definition file or simply just read in a new version of the data request and regenerate the data quickly.

In order to evaluate the performance of the PyConform tool, we chose to compare it against the performance of the software that we used for CMIP5. In this example we were limited to generating only fifty variables because this was the union of variables that matched between CMIP5 and CMIP6 for the atmosphere model.

In our evaluation on the Cheyenne supercomputer, we found that the original method took approximately 9 1/2 minutes to generate the CMIP compliant output and it took PyConform about 1 1/2 minutes to generate the same output using 16 MPI ranks. This provided us with over a six times speedup over existing methods. Since this was a smaller problem, we chose to run the timing tests on a smaller number of ranks. When PyConform was executed in a production mode for CMIP6, it generated thousands of variable files and we are able to scale out to more MPI ranks efficiently.

## 2.4  Data Publication

The final step in the CMIP workflow within Fig. 1 is publication of reviewed experiments to the ESGF, which is the data distribution and access platform designated for sharing CMIP and related simulation data. This part of the workflow was not automated. Instead it was a step triggered by the lead scientist once the data had been visually inspected. NCAR operates an ESGF Data Node, which is a software application stack that includes tools for checking conformance to the CMIP6 metadata standards, serving NetCDF data files using a Thredds Data Server and Globus Transfer, replication services, automated citation generation and experiment life cycle support, including data retraction and re-publication.

A significant challenge with CMIP5 data publication was managing the velocity and complexity of data publication using ad hoc communications, such as email. Given the challenges of post-processing noted above, each experiment was published incrementally. This led to multiple versions of experiments and added unnecessary complexity to the publication process. A separate challenge was managing an evolving ESGF software stack during the production CMIP5 data publication. The burden of updating the ESGF node frequently coupled with changing metadata requirements led to further slow-downs in the overall process.

For CMIP6 the ESGF software components were significantly improved, due to the increase in diversity, complexity and volumes being managed, as well as the experiences of data managers and node operators during CMIP5. In addition, a number of new components were developed for CMIP6, including the PrePARE data QC tools, a data replication tool, the Errata Service, and the Citation Service. These components were tested through a series of five "Data Challenges", which NCAR participated in as a member of the CMIP Data Node Operations Team (CDNOT)(Petrie et al., 2020) from January to June

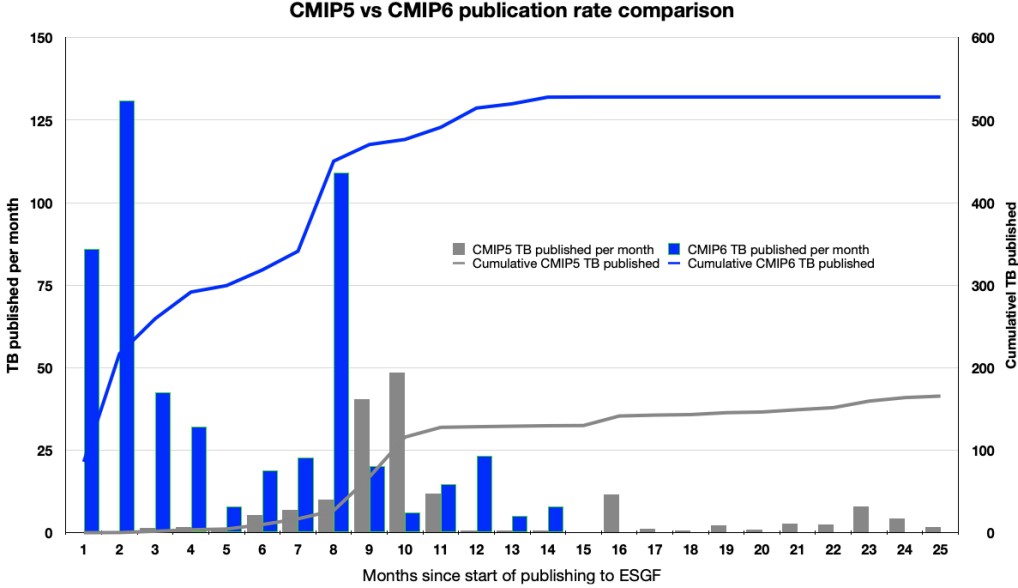

**Figure 8.** The cumulative and per month increases in the volume of data published to ESGF. During CMIP5, the ESGF software was stressed and problems arose. Despite those problems, NCAR was able to publish 165 TB of data. In preparation for CMIP6, the problems with the ESGF software stack were addressed and through these improvements, NCAR was able to publish 528 TB of data within 14 months, a three times increase in volume. In the first two months of the CMIP6 publication process, NCAR smoothly published over 216 TB of data, over 50 TB more than it contributed towards CMIP5.

2018. These data challenges were performed in advance of the model data availability and served to strengthen and improve the ESGF software stack with a series of integration and other system level tests. The significant improvements to the ESGF software stack and related tools vastly improved the rate of data publication for CMIP6. These performance improvements are shown within Fig. 8. In the first two months of the CMIP6 publication process, NCAR was able to smoothly publish 50 TB more than it had published in the full 25 months it took to publish data towards the CMIP5 campaign because of the

improvements that were made. This data (ESGF-NCAR, 2020; CMIP6 Data References, 2020) is available to download via ESGF.

Another reason why we were able to publish large volumes of data quickly is because we had used a stand alone version of PrePARE (PrePARE, 2020) to verify that our data contained all of the correct attributes before we started the publication process. The PrePARE package is part of the CMOR3 (Taylor et al., 2006) package produced by the Program for Climate Model

Diagnostics and Intercomparison (PCMDI) group within the Lawrence Livermore National Laboratory. The small problems that PrePARE was able to find allowed us to make corrections before large quantities of data were generated. Once the errors were corrected, PrePARE allowed us to verify that the data that was being created from the PyConform tool met the standards and gave us the confidence that we would be able to pass the publishing quality verification checks.

Another improvement to ESGF involved data versioning. Each ESGF data set is allocated a version number. This allows any data set to be uniquely referenced. Versioning enables data managers to retract any data that may have errors and replace it with a new version without any interruption on any ancillary services. This method of versioning allows all end users to know which data set version was used in their analysis, making data versioning critical for reproducibility.

ES-DOC was used to document climate models that participated in CMIP6 as well as to document the data sets the participating modeling centers published to ESGF (Pascoe et al., 2020). CESM2 has been extensively documented in ES-DOC (ES-DOC Model, 2020). Links to the unique ES-DOC documentation pages for each data set published are located within each NetCDF file within the CMIP6 collection on ESGF. The link can be located via the global history attribute `further_info_url`.

## 3    Process Workflow

During the completion of the CMIP5 simulations, each of the processes illustrated in Fig. 1 was an independent task, and each task was not automatically run in succession. Another problem was that each of the tasks were run by different individuals causing workflows to stop while they waited for someone to start the next task. For a run to have continuous forward progress, it needed to be monitored repeatedly at all hours and people needed to be on call continuously to post-process the data and this was not practical. There was also no fault-tolerance built into this workflow. If part of the simulation failed because of machine error, the simulation stopped and it wouldn't restart until someone did a manual check.

We adopted the use of Cylc (Oliver et al., 2018) for our CMIP6 experiments in order to coordinate the execution of all of the tasks within the end-to-end workflow of an experiment except for the publication task. Cylc is a workflow management tool developed at the National Institute of Water and Atmospheric Research (NIWA) and supported through NIWA and the UK Met Office. We also evaluated Rocoto (Harrop, 2017) as a workflow management tool. While Rocoto provided the basic functionality we required, we preferred Cylc's more robust interface and we valued its larger active development community.

A Cylc workflow can be invoked through command line tools or through a graphical user interface (GUI). Both provide intuitive control of the workflow and the individuals tasks. In order to track the status of all of the tasks within a workflow, Cylc updates its internal database that contains information about each of the tasks. This allowed the workflow to recover to a previous state if a problem was encountered on the machine.

The Cylc workflows were able to incorporate all of the tasks that a user wanted to execute. This included the model iterations, the moving of data, and the running of all of the Python tools discussed in this paper. The ability to automate the submission of all of the tasks we needed to run made the end-to-end workflow seamless and users did not have to worry about submitting any of the tasks by hand. This also eliminated the needed expertise to run the post-processing tools. Instead they were setup correctly and automatically ran as part of the workflow. All of these tasks are shown within an example Cylc dependency workflow graph within Appendix Fig. A1 and the Cylc workflow description file used to create this workflow can be found within the CESM-WF repository (Mickelson, 2020b).

Cylc also provided fault-tolerance within the workflows by allowing users to specify if they would like for Cylc to try rerunning a particular executable if it fails. For example, if one of the model runs failed because of machine error, it was

resubmitted to the queue and rerun without user intervention. This became extremely useful when compute nodes on Cheyenne would become unresponsive due to network issues. In these cases, the CESM execution would fail and the non-zero failure exit code triggered Cylc to resubmit the task again. This allowed us to automatically continue our workflow during the many network issues that plagued Cheyenne while we executed these simulations. Without this Cylc feature, we would of had to resubmit the tasks to the queue by hand and this would have caused us to loose productivity.

The process of setting up a CMIP6 workflow is complex because of the different tools that need to be setup for a particular experiment. This includes the setup of the Python tools discussed in this paper and the Cylc workflow description file. In order to reduce the burden on the users, a Python setup script (Mickelson, 2020a) performed many of the setup steps so users did not need any CMIP6 expertise. Once the users set the run time option values, such as run length and post-processing options, the script created the CESM experiment, created a post-processing directory, set up the post-processing tools for the specified CMIP6 experiment, and created a Cylc workflow definition file based on known task dependencies between the different tasks that were to run. After the script completed, users only needed to set experiment specific information, such as specific input file information and output variable names, into the CESM model and to build the model. Then the users started the experiment through Cylc. Human intervention was only needed if the Cheyenne login nodes went down or if the CESM simulation needed to be restarted from the beginning. In each case, users were able to restart the simulation from any point within the workflow. We have made our auto generated Cylc suite definition files that we used to run our CMIP6 experiments available on github (Mickelson et al., 2020).

Once the experiment was started with Cylc, the user running the simulation was able to view the simulation's progress through the GUI or command line interface. Users were also able to pause or stop individual tasks, rerun tasks, or skip tasks. It was also possible to add and remove tasks from the workflow graph after Cylc had started to execute. Cylc also provided process status information for all of the tasks, including start and stop run times and job identification numbers given by the queuing system. This provided our users with the control they needed to run the full experiment and any post-processing task.

None of our workflow users had any experience with the new Python tools we developed nor did they have any experience with Cylc before starting their first CMIP6 experiment. Therefore each of our users needed to be trained. We provided each of our twenty users an individual training session that lasted roughly two hours. We also answered questions they had via direct email and through an NCAR CMIP6 email group that was setup to only contain the workflow users and a few individuals from the NCAR supercomputing user support group. An addition to this support, we provided documentation (Mickelson, 2019b). The documentation walks through the workflow setup instructions, provides instructions on how to run Cylc, and provides answers to common questions we would receive. As a final method of support, we helped monitor the status of their simulations along with them and provided direct help when needed. We found that these training methods provided the confidence most of our users needed to finish their first simulation and to try setting up their next experiment independently.

The Cylc workflow description file was setup to run each task as a separate job in the PBS scheduler on Cheyenne. Therefore, each CESM model iteration and post-processing task that needed to be performed were separate jobs within the PBS scheduler and they were submitted only when the task the proceeded it finished successfully. Throughout the duration of our CMIP6 experiments, we were able to achieve high throughput of our experiments with low wait times in the PBS queue. As a result, we

did not suffer from lower job priorities by not pre-staging our jobs, though our approach may have achieved slower performance on busier systems that give higher priority to jobs that have been queued for longer periods of time. For running on these systems, it is possible to configure the Cylc workflow description file to submit multiple jobs to the queue at once with dependencies on each other in order to increase job throughput.

The modifications described within this section had the largest positive impact on our ability to complete our contributions towards CMIP6. Through this process, the users did not have to have expert knowledge on how to run any of the post-processing tools nor did they need to know how to format the published data. This eliminated the need to have an extra person run the post-processing tasks by hand. Also, having the workflow submit all tasks and resubmit failed tasks allowed us to complete experiments and publish data sooner because everything was continuously running. Finally, it made the process of completing complex experiments easier. As an example, for a particular MIP exercise, we were required to run eight different experiments, each containing one-hundred ensemble members (Deser and Sun, 2019; Smith et al., 2019). This would have required the user to build all eight-hundred experiments, run each experiment, create time-series files for each, and then create the standardized files for each experiment and all these steps would have been done by hand. This becomes a labor intensive process that requires extensive bookkeeping. With the workflow automation provided by Cylc, we were able to complete the eight experiments with each taking only a couple of days to complete and the user was only required to run a script that set up each case and to click on a start button.

## 4   Experiment Documentation

The experiments that NCAR had done for CMIP5 contained little documentation and no provenance was obtained. This made the simulations difficult to reproduce without having to contact the person who ran the original simulation. Another problem that was encountered was that it was difficult to track the progression of the simulations for CMIP5. During the process, only one individual knew which runs were in progress, the status of each of the simulations, and what was complete. To address these problems for CMIP6, the CESM experiment database was extended to provide the extra features that were needed.

The first task was to make it easier for the scientists to enter new experiments into the database. The previous version of the database required users to enter several pieces of information and this made it a cumbersome process. To improve this, known information was automatically harvested from the CESM experiment through CESM XML query commands and from the CMIP6 data request. This reduced the number of fields users had to fill in by hand and made the process more streamlined.

The next task was to allow for the experiments to upload the specific setup configuration files that were used and the CESM run-time timing files to a subversion repository so that experiment provenance could be captured. This was done by adding a subversion commit call right after a run iteration completed. When the data archiving step was run, all of the experiment's configuration and timing files were gathered up and were uploaded to a new svn subversion directory with the current date stamp. The database then gathered the differences and displayed them under the experiment's entry within the database. This allowed users to quickly identify changes that were made mid-run. Noting these changes and when they happen is critical to

reproducing an experiment. These changes can be scripted into the experiment's Cylc workflow definition file and this would allow it to be reran exactly how the original was run.

Another feature request was to link the diagnostic package results (CESM Diagnostics Results, 2019) to the database. As discussed in Section 2.2, the diagnostic packages produce several plots that are linked within an HTML document. The workflow uploaded these HTML documents automatically to a web server so people could view the results as soon as they were produced. The links to these web pages are found within each experiment's entry in the database so others could easily locate all of the results at one location.

The final feature request was to provide real-time experiment status within the database. As stated previously, it was difficult in CMIP5 to know the status of any given experiment. In order to update the run-time status within the database, we had each experiment's Cylc workflow email the database its status. This new interface allowed management and scientists to monitor the status of all CMIP6 experiments and to identify simulations that had stopped running and that had not finished.

Collectively these enhancements allowed us to track the progress of the experiments and document model configurations, output, and diagnostics all within one utility. This work also lead itself to other research projects that allowed for analyzing the timing information that was collected from each model run in order to study the model performance over time. This allowed for the identification of a degradation of performance after a machine upgrade, users selecting imbalanced processor layouts for their model runs, and model performance degradations (Mannik, 2019). This information can be then used to improve model performance and allow for more efficient use of computational resources.

## 5  Conclusions

Every generation of MIP exercises introduces new layers of complexity. We learned in CMIP5 that we could no longer use traditional serial methods to post-process the required amount of data and still meet our deadlines. CMIP6 required us to develop a new tool chain and forced us to change our methodologies. These new methods, described in this paper, provided us with an 18-times speedup. This allowed us to meet our deadlines and we were able to publish more than half a million data sets on the ESGF (ESGF-NCAR, 2020; CMIP6 Data References, 2020) for the CMIP6 project.

While Cylc has a learning curve, it was shown through this work to be extremely useful in coordinating all of the individual tasks of running a simulation, running diagnostics, and post-processing the data. It was shown to save both human time and time to simulation completion. Because of this success, Cylc is being more tightly integrated within CESM. This tighter integration now resides within the CESM infrastructure code and a Cylc workflow can now be generated with an option set within the CESM environment instead of it being a standalone Python script.

While we have shown that our new Python tools were successful, we believe these fundamental tasks should also be integrated more tightly within the CESM. This includes the time series and data conforming tasks. The current practices force multiple versions of the data to be on disk at a given time. As future MIP's grow more complex, their requested data volumes grow larger. This growth in data being requested makes it more difficult to carry multiple versions of the data around and the tighter integration of having the formatted data generated directly from the model simulation will allow us to save disk space.

The work we have done to improve the diagnostic packages has inspired new analysis workflows written by our scientists in Python. This work is being designed to run analytics on data that resides on our compute resources and on cloud platforms interchangeably. Current efforts are underway to combine these individual efforts in order to produce new versions of the diagnostic packages.

Having stronger data standards that the community conforms to will help ease the ability to perform intercomparisons across models. As more modeling centers move their data onto the cloud, the interest to compare results between models increases, and the community should make their data as easy to use as possible within these types of analytic workflows. We believe `dreqpy` is a great resource that moves the community towards that direction.

The complexity continues to grow with every generation of CMIP, and focused efforts are needed to coordinate the improvements to the infrastructure code around these attempts. While we present a detailed description of the workflow we chose to use for CMIP6, we hope to encourage other centers to evaluate their own workflows. It is important to consider developing flexibility within these types of workflows as workflow tools should be able to adapt to changes easily. Other important considerations when evaluating workflows include a reduction in the data footprint and an increase of the model data throughput. CMIP exercises are resource-expensive and time-consuming to run and it is important to be prepared for the commitment involved with these types of campaigns.

# Appendix A

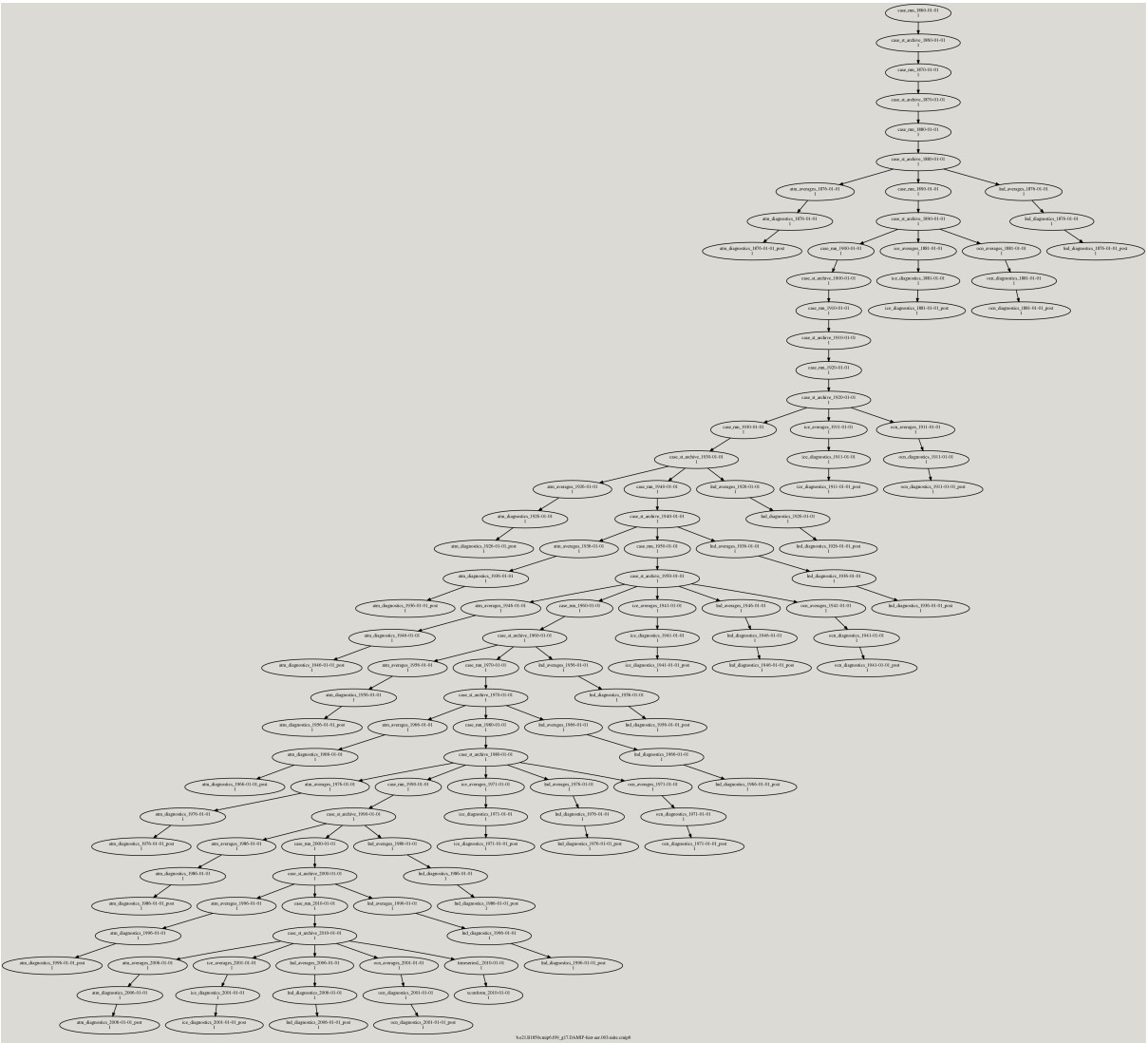

**Figure A1.** This figure shows a Cylc dependency workflow graph that was generated for an experiment we ran for CMIP6 which required us to simulate the climate from 1850 to 2015. The CESM model and its data movement utility were ran within this workflow. The CESM component diagnostic packages were also ran within this workflow. The ocean and ice model diagnostics were ran after every 30 years of simulation and the atmosphere and land model diagnostics were ran after every 10 years of simulation. The different diagnostic packages were ran as part of the many three chained tasks that are leaves within the workflow graph. The first task runs the PyAverager to generate the climatologies. The second task runs the NCL scripts in parallel to generate the plots and the web pages they are displayed on. The third task is a post command that copies the image and web files onto the web server. The PyReshaper and the PyConform tools were ran at the end of the simulation.

*Code availability.* The versions of the code that were used within our end to end workflow process for CMIP6 can be found at the following locations:

The version of the PyReshaper (version 1.0.6) that was used in this work can be downloaded from
        https://doi.org/10.5281/zenodo.3894842 (Paul et al., 2018). Further information can be found at https://github.com/NCAR/PyReshaper.
        The version of the PyAverager (version 0.9.16) that was used in this work can be downloaded from
        https://doi.org/10.5281/zenodo.3894862 (Mickelson et al., 2018). Further information can be found at https://github.com/NCAR/pyAverager.
        The version of the PyConform (version 0.2.8) that was used in this work can be downloaded from

https://doi.org/10.5281/zenodo.3895009 (Paul et al., 2019). Further information can be found at https://github.com/NCAR/PyConform.
        The version of the CESM post-processing framework (version 2.2.1) that was used in this work can be downloaded from
        https://doi.org/10.5281/zenodo.3895033 (Bertini and Mickelson, 2019). Further information can be found at
        https://github.com/NCAR/CESM_postprocessing.
        The version of the CESM workflow generation tool set (version 1.0) that was used in this work can be downloaded from

https://doi.org/10.5281/zenodo.3895058 (Mickelson, 2020a). Further information can be found at https://github.com/NCAR/CESM-WF.
        The CESM model (version 2) can be found at https://doi.org/10.5065/D67H1H0V (Danabasoglu et al., 2020). This work used the CESM
        versions
        2.1.0 (https://doi.org/10.5281/zenodo.3895306),
        2.1.1 (https://doi.org/10.5281/zenodo.3895315),

2.1.2 (https://doi.org/10.5281/zenodo.3895328).
        For this work we used Cylc version 7.8.3. The source code for this version can be retrieved at https://doi.org/10.5281/zenodo.3243691
        and it is referenced within Oliver et al. (2018).

*Author contributions.* SM led the development of the end-to-end CESM workflow for CMIP6. AB contributed to the development of the CESM post-processing framework and developed the CESM experiment database. GS managed the data for the project and was an advisor.

KP contributed to the development of the Python post-processing tools. EN published NCAR's CMIP6 data onto ESGF. JD and MV were advisors for the project.

*Competing interests.* The authors declare that they have no conflict of interest.

*Acknowledgements.* We thank Adam Phillips (NCAR) for his input and contributions towards the ES-DOC documentation. We thank the scientists at NCAR for their contributions towards the CEM model development and the running of the CMIP6 experiments. We also thank our

two anonymous referees for their time and helpful suggestions. The CESM project is supported primarily by the National Science Foundation (NSF). This material is based upon work supported by the National Center for Atmospheric Research, which is a major facility sponsored by the NSF under Cooperative Agreement No. 1852977. Computing and data storage resources, including the Cheyenne supercomputer (doi:10.5065/D6RX99HX), were provided by the Computational and Information Systems Laboratory (CISL) at NCAR.

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
