# Peer review of "A New End-to-End Workflow for the Community Earth System Model (version 2.0) for CMIP6"

_Geoscientific Model Development, 2020_

## Referee Comment (RC1) · Anonymous Referee #1 · 24 Jul 2020

This manuscript uses CMIP5 and 6 as a case study to show the improvements made to the workflow in order to support CMIP6 at NCAR/CESM. While this work is appreciated and does do a decent job of comparing the workflow from the CMIP5 era to the present, the information presented in the manuscript needs to be more efficiently conveyed so it's useful to the community as a whole. Information on CMIP5 workflow is not very elaborate to get an in-depth understanding and appreciation for the CMIP6 workflow efforts. Some parts of the manuscript can be much more than just an "internal documentation". Schematics like Figure 1 can be vastly improved. Information in the manuscript should serve as a motivation point for other labs to consider new workflow models. Several points are highlighted below. Most importantly, human time consumption could also be provided in this manuscript. Secondly, as we move towards

a new computing era, the manuscript should also let the readers know what else is out there in order to develop a new workflow, inspired by this manuscript. A data agnostic model, processing workflow, cloud-optimized workflows,etc should be touched upon at least in the conclusions section.

Overall, thanks for the manuscript and congratulations on publishing CMIP6 data on the ESGF.

Page 1 Ln 3: Statements such as "Many centers were not prepared.." needs to be rephrased to indicate the unexpected increase in complexity of CMIP6. Unless the relevant facts are cited and the data is provided, comments about other centers does not seem appropriate here.

Page 1 Ln 7: It is nice that there is six times improvement. Please verify if the actual data volume is specified in the manuscript for CMIP5 and CMIP6.

Page 1:Ln 20: For a diverse audience to follow, please indicate what is meant by post-processed data when it's mentioned for the first time.

Page 1. Ln 24. What is the factual evidence to show CESM ran relatively quickly compared to the other climate models? Are you referring to models from NCAR or other modeling centers? Is this number from a CPMIP computational metrics calculation? If there is no factual evidence or appropriate citation, this statement does not seem to be appropriate.

Page 2: Ln 1: What are the software inefficiencies? Workflow development tools? Please expand on this. Software seems very generic.

Page 2: Ln 44, The line that begins with "For CMIP5..", please break this sentence into two or three and avoid using "it.." several times in the same sentence, for clarity.

Page 2: Ln 45: What is different in CMIP6 versus CMIP5 in terms if requiring expert knowledge to ensure data met the correct standards? It does require expert knowledge in order to verify the correctness of scientific model output regardless for the phase

of the CMIPs. If this is incorrect, please clarify and rephrase the sentence to avoid confusion.

Page 2 Ln 46: How is the standardized data verified? What is the Quality Assurance process in the workflow? What is meant by "standardized data"? Please spell out the conventions to be adhered to.

Page 2, Ln 51: Where are the simulations run? Information regarding the computational environment is completely missing. What is the name of the compute system? Is/Was there a batch worlflow, job scheduler etc, etc.

Page 3. Ln 67: "The publication of CMIP5 data contributions to the ESGF was also a bottle neck within the data workflow" - This line needs more clarity to indicate what exactly is referred to as the bottle neck. Is ESGF the bottleneck? The tone of the sentence could be more constructive if a community developed federated framework is being criticized. Just as an example, thought not complete. Though there were performance issues with respect to the data publication onto the Earth System Grid Federation for a few reasons, the performance increased phenomenally due to .....

Page 4, Ln 91 TYPO: task based parallelism

Page 4, Ln 102: Try and avoid starting the sentence with "Because" .

Page 4, Ln 110: How dependent is the diagnostics framework on the supported languages you've described? What is the potential to expand the supported languages to say Python, Ruby, etc ?

Page 6, Ln 113: There is a mention of "this work". Please clarify what work this entails as part of this sentence for clarification. There is also a TYPO in "Specifically ..". Consider changing resulting to "results in" as you see fit.

Page 6, Ln 125: Was this a total re-write of the post-processing framework since CMIP5? Page 7: Figure 5- Y axis units missing. Please check all figures as well. Figure 5 caption indicates "46 seconds". Is this in sync with what is shown in the actual

figure?please verify.

Page 7 Ln 122. Figure 5 simply has info on pyAverager. If there is a single image that cross-compares the speedup time for the two tools,that'd be effective and in sync with the text in Ln 122.

Page 7, Ln 140. Paraphrase this sentence, especially "without any modifications". E.g. ..to allow seamless multi-model comparison based on uniform data standards to avoid less rewriting and error-prone transformations while doing so.. etc.

Page 7, Ln 142. "required" is a strong word here and not quite accurate. please paraphrase this. Not all modeling centers used CMOR even for CMIP5.

Page 8, Ln 154. How are changes in dreqPy incorporated in the workflow? Was there a fixed version? How were corrections in the requirements considered and incorporated?

Page 9 Ln 158. CMOR may also have a Python interface, please double check and then change this sentence as needed.

Page 10, Section 2.4. There is no mention of Data Quality Assurance which is extremely important though parts of the workflow may be automated. Please indicate the steps taken to quality control datasets. PrePARE could be accounted for metadata QA, but not quite for data and I am curious how that was incorporated. PrePARE also comes at a later stage when the heavy lifting of data processing and prep is almost complete. So, a bug revealed at a later stage may have its own cons. There was also a similar CMOR checker in some form available for CMIP5, though for CMIP6 it was more robust.

Page 10, Ln 175: Not sure if it's a typo - under-development versus under-developed. This sentence needs to revised either way. A constructive tone would be great.

Page 10, Ln 181: Please cite CDNOT paper that was recently submitted if you haven't already. Ruth et al. 2020

Page 10, Ln 183: Consider changing "harden" to strengthen or similar.

Page 10, Section 2.4. Versioning and ESDOC are two important components in data publication and the ESGF. These are not touched upon and would add immense value to the manuscript to include the process for these.

Page 10. A schematic for Cylc, e.g. Cylc dependency graph/dashboard would add value to the manuscript. Page 10. Ln 201, What is the internal DB implementation? How easy or difficult was it to get started with Cylc and is Cylc also used in other domains? Does the user have the ability to monitor the processes via Cylc and re-submit a job if needed?

Page 11 Ln 207: Sample configuration files from Cylc would be helpful. A section to explain how reproducibility is achieved in the workflow with a schematic and a case study– would be helpful.

Page 11, Ln 212: What are the setup steps? Is there an example of a definition file in the github repository references?

Page 11, Ln 218. How is troubleshooting and monitoring happening with Cycl and your workflow? Who manages that?

Page 11. Ln 225-227. This is nice. Was data publishing part of the automated pipeline? Please explain. In a fully automated workflow, what were the testing strategies, version control mechanisms, provenance capture mechanisms, etc? There is little mention about a couple of things, but more the better to make the manuscript stronger and reachable.

Page 11, Section 4, Ln 229: Again, please be constructive. Be specific as to what experiments you're referring to, what model, what modeling centre.

Page 11, Ln 2356:How was information automatically harvested from the CESM experiment?

Page 12, Ln 239: Please expand on what configurations and timing files mean here.

Page 12: Ln 241: What is the "code" that's referred to here? Please elaborate

Page 12: Ln 244: Are there sample analysis figures that could be provided? Are there any collaborative work on the diagnostic package that needs to be acknowledged? Is there scope for collaborative efforts since some of the diagnostic packages can be helpful to the community as a whole.

Page 12: Nice- HTML docs for viewing results.

Page 12, Ln 250 This seems to speak about monitoring capabilities, although the information provided here is not very useful for readers to learn from this work.

Page 12, Section 5. Line 260. By traditional tools, are serial tools referred to ? Since this seems to be the major difference in your workflow paradigm since CMIP5?

Line 263: Are there citations to the datasets referred to here? Any information on how CMIP6 citations were processed for CESM.

Conclusion should have future work as well, because what is considered traditional today will not hold good for the years to come. Lessons learned from CMIP6 exercise needs to be magnified in order to move towards cloud-optimized workflows and flexible APIs. The manuscript should give some food for thought to the readers. Examples to show if (if not) CMIP may be the only style of experiments that the workflow processes should be clarified.

―――――――――――――――――――

---

## Referee Comment (RC2) · Anonymous Referee #2 · 29 Jul 2020

General comments :

This paper presents the work carried out to completely modify the CESM's post-processing workflow. It's interesting and useful to get an overview of such a process, but I think some information are missing for the paper to serve as an example for other communities.

During my reading I would have liked to know more information on the Cheyenne super-computer. For example, do you have some restrictions on the storage (volume quota, inodes quota), is this supercomputer dedicated only for CMIP6 experiments ? Did you have some restricitions on you cpu allocation for post-treatment ?

For each part, I think it can be useful to have an information on the human time and

[Figure]

FTE necessaries to realize the tool from scratch to the production.

It's really a great job to have created this workflow that can be used by a "normal" user, and that avoids the problem of knowing CMIP data that only relies on a few people.

Specific comments :

Introduction

- lines 24 & 25 : Can you add a graph in order to visualize calcul and post-treatment performances for NCAR and other climate models

Data Workflow

- line 41 : "it was time consuming" : can you precise if you are talking about "human time" (find the script, launch it, check it etc.) or cpu time ?

- Line 63 : can you explicite "FTE" before to use it for the first time ? How did you make the FTE estimation for the implementation of XIOS and for the development of your own new tools ?

Time Series Generation :

- line 96 to 104 : Can you precise in the text how many Time-series (493) are created by your evaluation. why did you stop the test to 144 MPI ranks and don't test with more MPI ranks ? Did you try with 493 MPI ranks ? Can you explain how finally you make your choice for the MPI ranks repartition you will use, I imagine there is a reflexion between the human time (5 $\frac{1}{2}$ hours with you previous workflow and now 4 $\frac{1}{2}$ minutes), the total CPU time (4 $\frac{1}{2}$ minutes * 144 = 10,8 hours), and your cpu allocation on Cheyenne. (this specific comment is done also for the other parts of your workflow)

- Line 102 : did you try to improve the way you done the variables distribution on MPI ranks ?

- Figure 3 : can you add the "ideal speedup" line on it ?

Diagnostics

- line 117 to 122 : can you add information on how the choice of subcommunicators's number was done, and of the MPI rank distribution on each subcommunicator.

- Line 128 to 130 : can you explain on which criterion was done the climatologies distribution on MPI ranks ?

- Line 135 : can you re-run the experiment on 32 MPI ranks, to fixed the distribution problem.

- Figure 5 : can you add the "ideal speedup" line on it ?

Conforming Data to Meet Specifications

- line 147 : can you explain what you mean by "flexible interface" ?

- Line 148 : can you describe the "task-parallel approach" you choose to implement ?

- Lines 152 a 153 : how users that are not experts on CMIP6 (as it's tell several times in the paper for example lines 218 & 219) can know which functionalities need to be create ?

Data Publication

- As far as I know PrePARE will check the correpondance between output metadata and what is wait by CMIP6. But it will not check outputs quality (for example : no missing time step on a time-series). Can you present how you manage the quality control of your cmip6 outputs files ?

- What happen if PrePARE return problems on outputs cmip6 files ?

Process workflow

- can you explain if learning how to use Cylc was easy or not ? Can you estimate time and FTE necessaries for this implementation ?

- Did you hesitate with another software ?

- Maybe it can be useful to add a graphic showing how cylc is incorporated to your workflow, with the call tree of all your tools.

- Line 213 & 215 : I don't understand the difference between "the users set the default values" and "users only needed to set experiment specific information". And if it's "default values" why users need to modified them ?

- Is Cylc workflow can solve all errors ? Or is there a need for human intervention from time to time?

Experiment Documentation

- Line 229 : "The experiments that ... no provenance was obtained" : can you precise if it's only for NCAR simulations or for all groups's simulations ?

- Line 251 : can you precise how are managed "simulations that ran into problems" ?

Technical corrections

- Line 54 : it's finish by a "," instead of a "."

- Line 55 : "steps including;" need to be modified by "steps including:"

- Line 77 : "Instead the data", I'm not sure that you want to tell "instead", maybe "by consequences" or something like this.

- Line 91 : "this task base parallelism" need to be modified by "this task based parallelism"

- Line 187 : "CMIP6", I think you want to write "CMIP5"

- Line 200 : "in order keep track of the statues of all of the running tasks. In order to track the status of all of the tasks ...", maybe you can avoid to write two time "in order ... tasks"

---

## Author Comment (AC1) · 2 Sep 2020

Thank you for your thorough review of our paper. All of your thoughtful suggestions have been implemented and it is our belief that these suggestions have made it a stronger paper that will be more relevant to the community. Most of your suggestions were added within the same portions of the paper, but in some cases, the suggestions were addressed in other areas of the paper because of relevance.
* * *

---

## Author Comment (AC2) · 2 Sep 2020

Thank you for your thorough review of our paper and for the thoughtful suggestions that you made. We have included all of your suggestions within our revised version. It is our belief that these suggestions have made it a stronger paper that is more clear. Most of your suggestions were added within the same portions of the paper, but in some cases, the suggestions were addressed in other areas of the paper because of relevance.

---

## Author Response (AR1)

**A New End-to-End Workflow for the Community Earth System Model (version 2.0) for CMIP6**

Sheri Mickelson, Alice Bertini, Gary Strand, Kevin Paul, Eric Nienhouse, John Dennis, and Mariana Vertenstein

The National Center for Atmospheric Research, Boulder, CO, USA
 Correspondence: Sheri Mickelson (mickelso@ucar.edu)

We would like to thank our two reviewers for their thorough review and thoughtful suggestions to make a stronger paper.  We have included all of their comments below in blue and in italics We have included our responses after each comment.  The line numbers we reference refer to the line numbers found within this response.

**Reviewer #1:**

*This manuscript uses CMIP5 and 6 as a case study to show the improvements made to the workflow in order to support CMIP6 at NCAR/CESM. While this work is appreciated and does do a decent job of comparing the workflow from the CMIP5 era to the present, the information presented in the manuscript needs to be more efficiently conveyed so it's useful to the community as a whole. Information on CMIP5 workflow is not very elaborate to get an in-depth understanding and appreciation for the CMIP6 workflow efforts. Some parts of the manuscript can be much more than just an "internal documentation". Schematics like Figure 1 can be vastly improved. Information in the manuscript should serve as a motivation point for other labs to consider new workflow models. Several points are highlighted below. Most importantly, human time consumption could also be provided in this manuscript. Secondly, as we move towards a new computing era, the manuscript should also let the readers know what else is out there in order to develop a new workflow, inspired by this manuscript. A data agnostic model, processing workflow, cloud-optimized workflows, etc should be touched upon at least in the conclusions section. Overall, thanks for the manuscript and congratulations on publishing CMIP6 data on the ESGF.*

Figure 1 was improved by changing the orientation to project a more natural flow of the data.  The shapes were modified to reflect recommendations for flowcharts and colors were muted.  It was also added that this is a simplistic representation of our workflows and we point to a Cylc workflow graph within the Appendix to show an example of a flowgraph we use in practice.

We tried to address the amount of human time where appropriate.  We've added information about human time at lines 33, 56, 69, 72-74, and 326-334.

We addressed future analytic platforms within the conclusion and argued for the need of standardization of data to help make analysis easier on these types of platforms.

*1.Page 1 Ln 3: Statements such as "Many centers were not prepared.." needs to be rephrased to indicate the unexpected increase in complexity of CMIP6. Unless*

*the relevant facts are cited and the data is provided, comments about other centers does not seem appropriate here.*

This was a statement that was discussed informally amongst different centers that participated in CMIP5. This statement was removed (line 3) from the document as an appropriate citation does not exist.

*2. Page 1 Ln 7: It is nice that there is six times improvement. Please verify if the actual data volume is specified in the manuscript for CMIP5 and CMIP6.*
This was added starting at line 7 through line 10 within the Abstract.

*3. Page 1:Ln 20: For a diverse audience to follow, please indicate what is meant by postprocessed data when it's mentioned for the first time.*
This was addressed within the additions at line 23 and at line 48 within section 2.

*4. Page 1. Ln 24. What is the factual evidence to show CESM ran relatively quickly compared to the other climate models? Are you referring to models from NCAR or other modeling centers? Is this number from a CPMIP computational metrics calculation? If there is no factual evidence or appropriate citation, this statement does not seem to be appropriate.*
Again, this was a statement that was discussed informally amongst different centers that participated in CMIP5. This statement was removed from the document as an appropriate citation does not exist (lines 27-28).

*5. Page 2: Ln 1: What are the software inefficiencies? Workflow development tools? Please expand on this. Software seems very generic.*
This was modified at lines 28-29 to be more specific.

*6. Page 2: Ln 44, The line that begins with "For CMIP5..", please break this sentence into two or three and avoid using "it.." several times in the same sentence, for clarity.*
This sentence was modified starting at line 49.

*7. Page 2: Ln 45: What is different in CMIP6 versus CMIP5 in terms if requiring expert knowledge to ensure data met the correct standards? It does require expert knowledge in order to verify the correctness of scientific model output regardless for the phase of the CMIPs. If this is incorrect, please clarify and rephrase the sentence to avoid confusion.*

This statement is clarified within lines 50-52.

*8. Page 2 Ln 46: How is the standardized data verified? What is the Quality Assurance process in the workflow? What is meant by "standardized data"? Please spell out the conventions to be adhered to.*

The verification and quality assurance portion of this comment is addressed within the additions in line 52 and lines 238-239. The standardized data clarification is addressed within the additions in line 48.

9. *Page 2, Ln 51: Where are the simulations run? Information regarding the computational environment is completely missing. What is the name of the compute system? Is/Was there a batch workflow, job scheduler etc., etc.*

Information about our compute platforms was added at line 80 through line 86. Information about the scheduler was added within Process Workflow section (section 3) at line 335. These changes caused for the removal of the citation at line 116 and the additional clarification needed at line 231.

10. *Page 3. Ln 67: "The publication of CMIP5 data contributions to the ESGF was also a bottle neck within the data workflow" – This line needs more clarity to indicate what exactly is referred to as the bottle neck. Is ESGF the bottleneck? The tone of the sentence could be more constructive if a community developed federated framework is being criticized. Just as an example, thought not complete. Though there were performance issues with respect to the data publication onto the Earth System Grid Federation for a few reasons, the performance increased phenomenally due to …..*

This paragraph was modified as suggested. It can be found starting at line 75.

11. *Page 4, Ln 91 TYPO: task based parallelism*

This was changed at line 109. Thanks for catching that.

12. *Page 4, Ln 102: Try and avoid starting the sentence with "Because" .*

This was specifically addressed at line numbers 121 and 163.

13. *Page 4, Ln 110: How dependent is the diagnostics framework on the supported languages you've described? What is the potential to expand the supported languages to say Python, Ruby, etc. ?*

This line was modified in order to address this question and the response can be found at line numbers 136 through 140.

14. *Page 6, Ln 113: There is a mention of "this work". Please clarify what work this entails as part of this sentence for clarification. There is also a TYPO in "Specifically ..". Consider changing resulting to "results in" as you see fit.*

The clarification was added to lines 144-145 and, as a result, the typo was removed.

15. *Page 6, Ln 125: Was this a total re-write of the post-processing framework since CMIP5?*

A clarification was added to lines 162-163.

16. *Page 7 : Figure 5- Y axis units missing. Please check all figures as well. Figure 5 caption indicates "46 seconds". Is this in sync with what is shown in the actual figure? Please verify.*

Figure 5 is now Figure 6 and in it the Y axis was modified to be clearer that it's a speedup and the units are in "x".  Since it's in speedup vs. actual time, the 46 seconds was added as a reference point to give readers an idea on how much time a task takes to complete.  This time was verified and the plot correctly represents the data presented.

17.     *Page 7 Ln 122. Figure 5 simply has info on pyAverager. If there is a single image that cross-compares the speedup time for the two tools, that'd be effective and in sync with the text in Ln 122.*

Figure 5 was added to address this comment.  It shows a comparison that includes the time to create both the climatologies and all of the plots for three of the diagnostic packages.  The paragraph starting at line 166 was written to describe these results. The paragraph starting at line 171 was modified slightly to help with the flow within the descriptions of Figures 5 and 6.

18.     *Page 7, Ln 140. Paraphrase this sentence, especially "without any modifications". E.g. ..to allow seamless multi-model comparison based on uniform data standards to avoid less rewriting and error-prone transformations while doing so.. etc.*

This is clarified within lines 187-188.

19.     *Page 7, Ln 142. "required" is a strong word here and not quite accurate. Please paraphrase this. Not all modeling centers used CMOR even for CMIP5.*

This was modified in line 190.

20.     *Page 8, Ln 154. How are changes in dreqPy incorporated in the workflow? Was there a fixed version? How were corrections in the requirements considered and incorporated?*

This information was added into the new paragraph starting at line number 212.

21.     *Page 9 Ln 158. CMOR may also have a Python interface, please double check and then change this sentence as needed.*

We modified lines 192-193 to reflect that we are referring to our code.  CMOR does have a Python interface, but the problem resided within our software and not within CMOR.

22.     *Page 10, Section 2.4. There is no mention of Data Quality Assurance which is extremely important though parts of the workflow may be automated. Please indicate the steps taken to quality control datasets. PrePARE could be accounted for metadata QA, but not quite for data and I am curious how that was incorporated. PrePARE also comes at a later stage when the heavy lifting of data processing and prep is almost complete. So, a bug revealed at a later stage may*

*have its own cons. There was also a similar CMOR checker in some form available for CMIP5, though for CMIP6 it was more robust.*

We have added a paragraph within the Data Publication section (2.4) starting at line 260 that talks about how we used PrePARE to verify the correctness of our data. We also added clarification to lines 238-239 indicating that the data was also verified by scientists visually inspecting the data before it was hand triggered to be published.

23. *Page 10, Ln 175: Not sure if it's a typo – under-development versus under-developed. This sentence needs to revised either way. A constructive tone would be great.*

This was changed in line 246.

24. *Page 10, Ln 181: Please cite CDNOT paper that was recently submitted if you haven't already. Ruth et al. 2020*

Thank you for bringing this to our attention. It is now cited at line 253.

25. *Page 10, Ln 183: Consider changing "harden" to strengthen or similar.*

This was modified at line 254.

26. *Page 10, Section 2.4. Versioning and ESDOC are two important components in data publication and the ESGF. These are not touched upon and would add immense value to the manuscript to include the process for these.*

Both of these concepts have been added within the two new paragraphs that are found within lines 267-274.

27. *Page 10. A schematic for Cylc, e.g. Cylc dependency graph/dashboard would add value to the manuscript.*

A Cyc workflow graph has been added as Appendix A1 and a citation has been included for where you can find the Cylc workflow code that was used to create that graph can be found. This is referenced from lines 297-299. The dashboard information can be found within the workflow documentation that is referenced within Mickelson, 2019b.

28. *Page 10. Ln 201, What is the internal DB implementation? How easy or difficult was it to get started with Cylc and is Cylc also used in other domains? Does the user have the ability to monitor the processes via Cylc and resubmit a job if needed?*

This information was added within the paragraph that starts at line 320. Information is also provided at lines 288-290. We also added training information that starts at line 326 that gives incite on how we made Cylc easier for our users.

29. *Page 11 Ln 207: Sample configuration files from Cylc would be helpful. A section to explain how reproducibility is achieved in the workflow with a schematic and a case study– would be helpful.*

A sample configuration file is cited at line 299. We have also published all of our workflow configurations in Mickelson, 2020 (see lines 318-319). Reproducibility is

further clarified with an addition within the Experiment Documentation section (section 4) at lines 372-374.

30.    *Page 11, Ln 212: What are the setup steps? Is there an example of a definition file in the GitHub repository references?*

This is clarified with the additions to lines 311-312 and lines 314-315. As stated above, information on where to find example definition files can be found within Mickelson, 2020 (see lines 318-319).

31.    *Page 11, Ln 218. How is troubleshooting and monitoring happening with Cylc and your workflow? Who manages that?*

Information in regards to Cylc automatically monitoring the status and resubmitting can be found at lines 303-307. Wording was modified in lines 300-302 to make that addition flow better. Monitoring by users is addressed though lines addition 320-321. Monitoring is also touched upon in the paragraph about training in lines 332-333.

32.    *Page 11. Ln 225-227. This is nice. Was data publishing part of the automated pipeline? Please explain. In a fully automated workflow, what were the testing strategies, version control mechanisms, provenance capture mechanisms, etc.? There is little mention about a couple of things, but more the better to make the manuscript stronger and reachable.*

Publication was not part of the automated workflow and this is now specifically stated at line 284 and at lines 238-239. Publication was triggered through a manual step within our database because we wanted to ensure that at least one person reviewed each simulation. This step was an authorization step verifying the correctness of the data.

33.    *Page 11, Section 4, Ln 229: Again, please be constructive. Be specific as to what experiments you're referring to, what model, what modeling Centre.*

This was clarified at line 356.

34.    *Page 11, Ln 2356:How was information automatically harvested from the CESM experiment?*

This information was added at lines 363-364.

35.    *Page 12, Ln 239: Please expand on what configurations and timing files mean here.*

This line was changed to add more clarity. The changes can be found at lines 366-367.

36.    *Page 12: Ln 241: What is the "code" that's referred to here? Please elaborate*

This is clarified at lines 369-370.

37.    *Page 12: Ln 244: Are there sample analysis figures that could be provided? Are there any collaborative work on the diagnostic package that needs to be acknowledged? Is there scope for collaborative efforts since some of the diagnostic packages can be helpful to the community as a whole.*

A citation was added to give an example of diagnostics we created for some of our PMIP4 experiments. This can be found at line 457 within the references and it is cited at lines 141 and 375. These packages are currently being deprecated and new versions are being created as suggested at lines 407-410 within the conclusion. These changes fall within our work towards Pangeo (https://pangeo.io/) and collaboration details can be found through the link to the Pangeo project.

*38.     Page 12: Nice- HTML docs for viewing results.*
Thank you

*39.     Page 12, Ln 250 This seems to speak about monitoring capabilities, although the information provided here is not very useful for readers to learn from this work.*
This line was removed as it was too detailed and did not add to the text (line 382-383). Lines 380 and 381 were modified as well.

*40.      Page 12, Section 5. Line 260. By traditional tools, are serial tools referred to ? Since this seems to be the major difference in your workflow paradigm since CMIP5?*
Yes, this was clarified in line 393.

*41.     Line 263: Are there citations to the datasets referred to here? Any information on how CMIP6 citations were processed for CESM.*
We have included two references at lines 259 and 396. The first is the location to download the data. The second contains the data citations that are automatically generated from the publication process.

*Conclusion should have future work as well, because what is considered traditional today will not hold good for the years to come. Lessons learned from CMIP6 exercise needs to be magnified in order to move towards cloud-optimized workflows and flexible APIs. The manuscript should give some food for thought to the readers. Examples to show if (if not) CMIP may be the only style of experiments that the workflow processes should be clarified.*
The conclusions section now contains some insights into cloud data analytic workflows. This is included within lines 408-409 and 411-414. The last paragraph of the conclusion was also modified to give readers a more precise take away message (lines 415-422). This includes the underlying themes that motivated us to design our tools the way we did and why evaluations and redesigns are necessary.

**Reviewer #2:**
*This paper presents the work carried out to completely modify the CESM's postprocessing workflow. It's interesting and useful to get an overview of such a process, but I think some information are missing for the paper to serve as an example for other communities. During my reading I would have liked to know more information on the Cheyenne supercomputer. For example, do you have some restrictions on the storage (volume quota, inodes quota), is this supercomputer dedicated only for CMIP6*

*experiments ? Did you have some restrictions on you CPU allocation for post-treatment ? For each part, I think it can be useful to have an information on the human time and FTE necessaries to realize the tool from scratch to the production. It's really a great job to have created this workflow that can be used by a "normal" user, and that avoids the problem of knowing CMIP data that only relies on a few people.*

We have added lines 10-13 and 415-422 in hopes to clarify the motivation for this paper and to help frame our intentions for the community as they read this paper.

We have added lines 80-86 to describe our compute platforms and lines 335-342 to talk about our queueing system.

We have not included details on our volume quotas, but CISL was generous and gave us enough disk space that our experiments were able to run on our system without running out of space. We developed a specific data plan that influenced where the workflow would create files.  For example, the raw files created directly from the model where put within our 1 PB scratch allocation and it was allowed to be purged off.  Our post-process files that we kept were put into a more permanent space.  This space was purchased specifically for CMIP6 and we determined the space we needed based on the CMIP6 data request for the experiments we were committed to.

We have also included information on human cost where appropriate and noted in your comments and from the comments of the reviewer #1.

We've added information about human time at lines 33, 56, 69, 72-74, and 326-334.

1. *Introduction – lines 24 & 25 : Can you add a graph in order to visualize calcul and post-treatment performances for NCAR and other climate models Data Workflow.*

This was a statement that was discussed informally amongst different centers that participated in CMIP5.  This statement was removed (lines 27-28) from the document as an appropriate citation does not exist.

2. *Line 41 : "it was time consuming" : can you precise if you are talking about "human time" (find the script, launch it, check it etc.) or CPU time ? -*

This is clarified within lines 44-45.

3. *Line 63 : can you explicite "FTE" before to use it for the first time ? How did you make the FTE estimation for the implementation of XIOS and for the development of your own new tools ?*

This is clarified within line 69.

4. *line 96 to 104 : Can you precise in the text how many Time-series (493) are created by your evaluation. Why did you stop the test to 144 MPI ranks and don't test with more MPI ranks ? Did you try with 493 MPI ranks ? Can you explain how finally you make your choice for the MPI ranks repartition you will use, I imagine there is a reflexion between the human time (5 1 2 hours with you previous workflow and now 4 1 2 minutes), the total CPU time (4 1 2 minutes \* 144 = 10,8 hours), and your CPU allocation on Cheyenne. (this specific comment is done also for the other parts of your workflow)*

This is now explained starting at line 124 through 127.

5. *Line 102 : did you try to improve the way you done the variables distribution on MPI ranks ?*

We did not, but an explanation is now provided at lines 128-132.

6. *Figure 3 : can you add the "ideal speedup" line on it ?*

This plot was modified to contain an ideal speedup line and it was changed to a line plot instead of a bar plot to show this information better.

7. *line 117 to 122 : can you add information on how the choice of subcommunicators's number was done, and of the MPI rank distribution on each subcommunicator. -*

This information was added starting at line 155 though line 159.

8. *Line 128 to 130 : can you explain on which criterion was done the climatologies distribution on MPI ranks ? -*

This information was added within the same paragraph as about at lines 158-159.

9. *Line 135 : can you re-run the experiment on 32 MPI ranks, to fixed the distribution problem. -*

This suggestion is addressed within lines 179-183.

10. *Figure 5 : can you add the "ideal speedup" line on it ?*

This plot was modified to contain an ideal speedup line and it was changed to a line plot instead of a bar plot to show this information better. For reference, this plot is now Figure 6 because another figure was suggested from reviewer #1.

11. *line 147 : can you explain what you mean by "flexible interface" ? -*

This line was specifically modified at lines 193-197 to add clarity. We also added two paragraphs that addresses the flexibility starting at line 217 though 227.

12. *Line 148 : can you describe the "task-parallel approach" you choose to implement ? -*

This line was removed at lines 199-200 and it was moved to lines 206-207 with a better explanation.

13. *Lines 152 a 153 : how users that are not experts on CMIP6 (as it's tell several times in the paper for example lines 218 & 219) can know which functionalities need to be create ?*

Through our process we had scientists define these for us. We listed the CMOR variables names along with the descriptions, requested units, and grids and they would note how CESM data would be used to derive the new CMIP6 variables. Their answers were then turned into a text file. An example of a definition text file can be found within the citation found within lines 203-204.

14. *Data Publication – As far as I know PrePARE will check the correspondence between output metadata and what is wait by CMIP6. But it will not check outputs quality (for example : no missing time step on a time-series). Can you present how you manage the quality control of your cmip6 outputs files ? – What happen if PrePARE return problems on outputs cmip6 files ?*

This is now addressed in several places throughout the paper. We describe how we used PrePARE within lines 260-266. We now address time step verification within lines 94-96. We also address quality control within lines 238-239. If problems were found, our procedure is now briefly described within lines 225-227.

15. *Process workflow – can you explain if learning how to use Cylc was easy or not ? Can you estimate time and FTE necessaries for this implementation ?Did you hesitate with another software ? – Maybe it can be useful to add a graphic showing how Cylc is incorporated to your workflow, with the call tree of all your tools. -*

We now address training on our workflow at lines 326-334. We address another software option (Rocoto) at lines 286-287. We also included a workflow graph (or call tree) within Appendix A1.

16. *Line 213 & 215 : I don't understand the difference between "the users set the default values" and "users only needed to set experiment specific information". And if it's "default values" why users need to modified them ? -*

This is clarified within lines 311-312 and lines 314-315.

17. *Is Cylc workflow can solve all errors ? Or is there a need for human intervention from time to time?*

Not all and some information about this at lines 316-319.

18. *Line 229 : "The experiments that . . . no provenance was obtained" : can you precise if it's only for NCAR simulations or for all groups' simulations ? –*

This is now clarified at line 356 to indicate that it was NCAR that did not have a formal way to record for provenance.

19. *Line 251 : can you precise how are managed "simulations that ran into problems" ?*

This was clarified at line 384.

**Technical corrections**

1. *Line 54 : it's finish by a "," instead of a "." -*

This is corrected at line 60.

2. *Line 55 : "steps including;" need to be modified by "steps including:" –*

This is corrected at line 61.

3.  *Line 77 : "Instead the data", I'm not sure that you want to tell "instead", maybe "by consequences" or something like this. -*

This is corrected at line number 92.

4.  *Line 91 : "this task base parallelism" need to be modified by "this task based parallelism"*

This is corrected at line number 109.

5.  *Line 187 : "CMIP6", I think you want to write "CMIP5" -*

This is corrected at line number 258.

6.  *Line 200 : "in order keep track of the statues of all of the running tasks. In order to track the status of all of the tasks …", maybe you can avoid to write two time "in order . . . tasks"*

The first sentence was re-written and it can be found at lines 288-290.

**Author's Changes**

The authors have made the following modifications:

Line 2: large was changed to dramatic in order to emphasize the increase in complexity.

Line 19: run was changed to experiments to be consistent with the language used throughout the paper.

Line 22:  the sentence was split in order to accommodate for the extra text that was requested.

Line 60:  Grammar fix.

Line 186:  Change was needed in order to match the label that was used in Figure 1.

Text in Figure 8:  Text was added for clarification purposes.

Line 294-295:  Clarify what "This" refers to.

[revised manuscript text omitted]

---

## Author Response (AR2)

**A New End-to-End Workflow for the Community Earth System Model (version 2.0) for CMIP6**

Sheri Mickelson, Alice Bertini, Gary Strand, Kevin Paul, Eric Nienhouse, John Dennis, and Mariana Vertenstein
The National Center for Atmospheric Research, Boulder, CO, USA
Correspondence: Sheri Mickelson (mickelso@ucar.edu)

• L. 48. Consider putting the sentence at the past tense: "After the standardized data were verified by the scientist, it was …"
Modified as suggested.

• L. 61 and L62. Consider putting a comma after "closed" and after "execution"
Modified as suggested.

• L.62. Consider changing "open and close" for "openings and closings"
Modified as suggested.

• L. 79-80-81: Consider putting this paragraph at the present tense.
Modified as suggested.

• L. 116: Consider splitting the sentence into two: "… between variables. Therefore, some ranks …"
Modified as suggested.

• Caption for Figure 3: Consider changing "existing" for "previous CMIP5 sequential"
Modified as suggested.

• L 128- 134: Is this valid for CMIP5 and CMIP6? Can you specify?
The first sentence was modified to add clarity.

• L. 134: can you give the more precise web reference?
Modified to add the landing page for all diagnostics and specific links to the four different packages for one of our CMIP6/PMIP4 experiments.

• Figure 4 captions: maybe precise that the figure is for 3 MPI ranks?
The caption was modified for clarity.

• L. 147: the first part of the sentence "The number of MPI ranks within a subcommunicator was set to four, unless the total number of ranks was less than four …" does not make much sense to me, as it is obvious. Maybe rephrase?
The first two sentences were modified to add clarity.

• L. 201-205: can you clarify how much of this is automated?

This section was was modified to add clarity.

• L. 265 and 266: don't use the plural form "were" when you write "each of"
Modified as suggested.

• L. 313: change "a hands" for "a hand"
Removed hands and modified the sentence slightly.

• L. 324: consider merging the 2 sentences: "… our jobs, though our …"
Modified as suggested.

• L. 345: consider adding "for CMIP6" after "these problems"
Modified as suggested.

• L. 364: consider adding "in CMIP5" before "to know"
Modified as suggested.

Author Modifications (line numbers refer to the difference document):

Line 124: changed "master-slave" to "coordinator-worker" for more inclusive language.

Change "Figure" to "Fig." For lines:
L. 39
Figure 1 caption
L. 85
L. 101
L. 130
L. 145
L. 170
L. 183
L. 194
L. 232
L. 252
L. 274
L. 294

L. 440-442: More Acknowledgements were added

[revised manuscript text omitted]

---

## Author Response (AR3)

**A New End-to-End Workflow for the Community Earth System Model (version 2.0) for CMIP6**

Sheri Mickelson, Alice Bertini, Gary Strand, Kevin Paul, Eric Nienhouse, John Dennis, and Mariana Vertenstein

The National Center for Atmospheric Research, Boulder, CO, USA Correspondence: Sheri Mickelson (mickelso@ucar.edu)

The authors have made two changes to the manuscript:

L. 470: Removed the extra 'h' to fix the broken link.

L. 536: Fixed the broken doi link.  The last digit ('2') had been ommitted.

[revised manuscript text omitted]